# A Biological and Immunological Characterization of *Schistosoma Japonicum* Heat Shock Proteins 40 and 90α

**DOI:** 10.3390/ijms21114034

**Published:** 2020-06-04

**Authors:** Zhipeng Xu, Minjun Ji, Chen Li, Xiaofeng Du, Wei Hu, Donald Peter McManus, Hong You

**Affiliations:** 1Department of Immunology, Molecular Parasitology Laboratory, QIMR Berghofer Medical Research Institute, Brisbane, QLD 4006, Australia; zhipengxu@njmu.edu.cn (Z.X.); Xiaofeng.Du@qimrberghofer.edu.au (X.D.); 2Department of Pathogen Biology, Nanjing Medical University, Nanjing 211166, China; jiminjun@njmu.edu.cn (M.J.); lichen262628@163.com (C.L.); 3Collaborative Innovation Center for Genetics and Development, State Key Laboratory of Genetic Engineering and Ministry of Education Key Laboratory of Contemporary Anthropology, School of Life Sciences, Fudan University, Shanghai 200433, China; huw@fudan.edu.cn

**Keywords:** heat shock protein, *Schistosoma japonicum*, granuloma, Th1, Th2, Th17

## Abstract

We characterized *Schistosoma japonicum* HSP40 (Sjp40) and HSP90α (Sjp90α) in this study. Western blot analysis revealed both are present in soluble egg antigens and egg secretory proteins, implicating them in triggering the host immune response after secretion from eggs into host tissues. These observations were confirmed by immunolocalization showing both HSPs are located in the Reynolds’ layer within mature eggs, suggesting they are secreted by miracidia and accumulate between the envelope and the eggshell. Both HSPs are present in the musculature and parenchyma of adult males and in the vitelline cells of females; only Sjp90α is present on the tegument of adults. Sjp40 was able to enhance the expression of macrophages, dendritic cells, and eosinophilic cells in mouse liver non-parenchymal cells, whereas rSjp90α only stimulated the expression of dendritic cells. T helper 1 (Th1), Th2, and Th17 responses were increased upon rSjp40 stimulation in vitro, but rSjp90 only stimulated an increased Th17 response. Sjp40 has an important role in reducing the expression of fibrogenic gene markers in hepatic stellate cells in vitro. Overall, these findings provide new information on HSPs in *S. japonicum,* improving our understanding of the pathological roles they play in their interaction with host immune cells.

## 1. Introduction

Schistosomiasis, caused by *Schistosoma japonicum*, is still a major public health problem that threatens many millions of people in the People’s Republic of China [1,2]. The most serious pathology of this disease is caused by egg-induced hepatic granulomas and fibrosis in the livers and intestines, and this impacts on the hosts’ quality of life and health status, resulting in severe morbidity and even mortality [3]. The pathological mechanism of hepatic granuloma formation is a soluble egg antigen (SEA) and egg secretory protein (ESP)-induced complex immune response that results in the accumulation of macrophages, eosinophils, dendritic cells, CD4^+^ T cells and other immune cells [4]. Determining how these various immune cells are activated and recruited within the liver environment will provide new avenues to further understanding the process of immunopathology in schistosomiasis. 

Heat-shock proteins (HSPs) constitute the first line of protection for cells exposed to stressful conditions. They constitute a large group of molecular chaperones, which have been classified into several families, including HSP27, HSP40, HSP60, HSP70, HSP90, and HSP110, based on their molecular weight and sequence homology [5,6]. Secretion of HSPs from *S. japonicum* eggs has been shown to induce strong immunomodulatory effects, including immunostimulatory and immunosuppressive reactivities [7,8]. One of the most abundant egg proteins from *S. japonicum* is the egg-derived heat shock protein 40 (HSP40, Sjp40), which is able to restrain the activation of hepatic stellate cells (HSCs) through the STAT3/p53/p21 pathway [9]. Evidence showing Sjp40 is secreted into blood at the early stage of infection suggested a potential diagnostic application [10]. In addition, our previous work showed that Sjp40 significantly induced the expression of co-stimulatory molecules (CD40, CD80, and CD86) and MHCII on the surface of the macrophages [11], suggesting an important role in the activation of antigen-presenting cells. Notably, the peptides p6 (51–70), p25 (241–260), and p30 (291–310) derived from Sjp40 were able to inhibit allergic asthmatic reactions through inducing IFN-γ production [12], indicating a novel form of immune protection through helminth infection.

Another abundant HSP derived from *S. japonicum*, named heat shock protein 90 (Sjp90), has been identified in adult worms [13]. Expression of Sjp90 was downregulated in UV-irradiated cercariae [14], probably due to the fact that the synthesis of heat shock proteins is inhibited by UV irradiation, which causes the parasite to release a variety of proteins with abnormal conformations [15]. In addition, proteome analysis showed Sjp90α (the inducible cytosolic isoform of Hsp90) is present at high concentrations in the excretory/secretory (ES) proteins of adult *S. japonicum*, implying a central role in immunomodulation in the host–parasite relationship [7]. Furthermore, an immunoproteomic study showed tegumental expression of Sjp90α and the HSP was predicted to interact with another tegument protein, a putative stress inducible protein 1 (STIP1, SJCHGC06661), which is highly antigenic and immunogenic and has six TPR domains and one STI1 domain [16]. However, the precise biological function of Sjp90α in schistosomes and its potential immunological role in the definitive host remain unclear and need to be determined.

In this study, we cloned and characterized Sjp40 and Sjp90α from *S. japonicum*. In addition, we used real-time PCR to measure the transcription levels of Sjp40 and Sjp90α in different life cycle stages. Immunolocalization studies of Sjp40 and Sjp90α were also undertaken in the eggs and worms. Furthermore, the potential effects of Sjp40 and Sjp90α on inflammatory cells, including macrophages, neutrophils, eosinophils, and T cells subsets, were analyzed by flow cytometry. 

## 2. Results

### 2.1. Determination of the Sequences and Tertiary Structures of Sjp40 and Sjp90α 

The sequence of *S. japonicum* HSP40 (Sjp40, ACL79582.1) comprises an open reading frame (ORF) of 1065 bp encoding 354 amino acids. Multiple alignments showed that Sjp40 shares 69% amino acid sequence identity with *Schistosoma mansoni* HSP40 (Smp_302290), 64% amino acid sequence identity with *S. haematobium*, and 31% amino acid identity with *Homo sapiens*. *S. japonicum* HSP90α (Sjp90α, CAX70123.1) comprises an open reading frame (ORF) of 780 bp encoding 259 amino acids. Sjp90α shares 82% amino acid sequence identity with *S. mansoni* HSP90 (Smp_072330), 43% amino acid sequence identity with *S. haematobium*, and 61% amino acid identity with *Homo sapiens*. 

The three-dimensional (3D) structure of the two proteins was predicted by homology modeling using the online SWISS-MODEL tool [17]. For the Sjp40 structure, we found a protein homolog with an alignment length of 338 amino acids and corresponding PDB ID: 2bol.1.A, and this is shown in Figure 1a,b; the Sjp40 protein and this homolog showed 32.69% identity, with 88% coverage. Additionally, it presents a high global model quality estimation score (GMQE: 0.61) evaluated using the Ramachandran plot method, showing that 91.07% of residues fall in the favorable region and 8.93% in the additional allowed region. With the full length Sjp90 structure (Figure 1c,d), we selected the most homologous protein with an alignment length of 719 amino acids and corresponding PDB ID: 5uls.1.A; this homolog showed 44.79% identity and 88% coverage with the Sjp90 protein. This template showed a high GMQE score (0.68) and a Ramachandran plot with 91.03% Sjp90 residues falling in the favorable region and 8.97% residues in the outlier region. The Ramachandran plot analysis show that the modeled 3D structures of Sjp40 and Sjp90 have acceptable stability (>90% in the most favored regions) following the rule of stereochemistry.

Conserved functional domains of Sjp40 and Sjp90α were identified by comparing sequence identity with HSPs of other species (Appendix A), including *S. mansoni, Homo sapiens*, and *Mus musculus*. Using the Scan Prosite of ExPASy analysis tools (https://prosite.expasy.org), 23 motif hits (by six distinct patterns) were identified in Sjp40 (Figure 1e) including (1) four N-myristoylation sites, which can acylate eukaryotic proteins and target proteins to cellular membranes [18]; (2) four protein kinase C phosphorylation sites, which exhibit a preference for the phosphorylation of serine or threonine residues found close to a C-terminal basic residue [19]; (3) six casein kinase II phosphorylation sites, whose activity is independent of cyclic nucleotides and calcium [20]; (4) three N-glycosylation sites, which relate to tyrosinase activity by helping protein folding [21]; (5) four cAMP- and cGMP-dependent protein kinase phosphorylation sites that contribute to the phosphorylation of serine or threonine residues [22,23]; and (6) two amidation sites. In the full length amino acid sequence of Sjp90α, there were 38 hits (by eight distinct patterns) presented (Figure 1f) including (1) seven protein kinase C phosphorylation sites; (2) twelve casein kinase II phosphorylation sites; (3) four N-glycosylation sites; (4) six N-myristoylation site; (5) four tyrosine kinase phosphorylation sites, which modulate enzymatic activity and recruit downstream insulin receptor substrate (IRS) proteins [24]; (6) three cAMP- and cGMP-dependent protein kinase phosphorylation sites; (7) two amidation sites; (8) a MEEVD motif, which is reported to provide the binding site for interaction with the tetratricopeptide repeat (TPR)-domain (mediate protein-protein interaction) containing co-chaperones [25,26]; and (9) 19 ATP-binding sites, which are extremely conserved as these sites are present in murine heat shock protein HSP90 (NP_032328.2) and indicate that Sjp90α has ATPase activity.

### 2.2. Stage-Specific mRNA Expression of Sjp40 and Sjp90α

To determine the transcription levels of Sjp40 and Sjp90α in different life cycle stages of *S. japonicum*, RT-PCR was performed and normalized with the house keeping gene PSMD4 (26S proteasome non-ATPase regulatory subunit 4), which has been validated as a standard reference gene in transcriptomic analysis of *S. japonicum* in a number of studies [27,28,29,30]. The analysis revealed that both Sjp40 and Sjp90α were expressed in schistosomula, adult worms, and eggs of *S. japonicum* (Figure 2a,b) with the lowest expression level recorded in cercariae. The transcription level of Sjp40 was strikingly higher in eggs than other stages (*F*_(4, 29)_ = 47.47, Least Significant Difference (LSD) post-hoc test: *p* < 0.0001) (Figure 2a), whereas Sjp90α expression was elevated in adult males with transcription levels in eggs, schistosomulum, and adult females being similar (*F*_(4, 29)_ = 124.1, all LSD: *p* < 0.0001) (Figure 2b).

### 2.3. Distribution of Sjp40 and Sjp90α in S. Japonicum Eggs

Immunolocalization of eggs trapped in infected mouse livers using either HRP labeling or immunofluorescence showed that native Sjp40 and Sjp90α were expressed inside immature eggs (Figure 3a), with both HSPs mainly localized to neural mass (NM) cells (one single large cell with numerous peripheral nuclei) and in the epidermis (EPI) cells of the intra-ovular miracidium within mature eggs (Figure 3b). Furthermore, fluorescence signaling of Sjp40 and Sjp90α was detected in the Reynolds’ layer (Figure 3b), which only appears between the envelope and the eggshell in the mature egg and may represent its accumulated secretions [31].

To determine whether Sjp40 and Sjp90α could be secreted by the *S. japonicum* eggs, Western blot analysis was undertaken using mouse anti-rSjp40 or rSjp90α serum to probe egg secreted proteins (ESP) and soluble egg antigens (SEA) from purified eggs (Appendix A). As shown in Figure 3c, native Sjp40 was recognized by anti-rSjp40 serum both in SEA and ESP at the predicted size of 39.2 kDa. A clear band at approximately 100 kDa was recognized by anti-rSj90α both in SEA and ESP (Figure 3d); this was probably due to post-translational modification of the full length Sjp90, due to glycosylation and/or phosphorylation [32], but this needs to be further investigated. Together, these results indicate that Sjp40 and Sjp90α are secreted by the mature eggs of *S. japonicum*.

### 2.4. Immunolocalization of Sjp40 and Sjp90α in Adult S. Japonicum

We determined the distribution of Sjp40 and Sjp90α in male and female worms using both HRP labeling and immunofluorescence; the results showed that native Sjp40 is not only expressed in the underlying musculature but is also present throughout the parenchyma of males and the vitelline cells of females (Figure 4 and Appendix A). Sjp90α was expressed throughout the parenchyma and testis of males and vitelline cells of female; In addition, the Sjp90α is present in the tegument of male and female worms (Figure 4 and Appendix A). 

### 2.5. Effect of rSjp40 and rSjp90α on Hepatic Immune Cells 

To better understand the immune roles of the egg-secreted Sjp40 and Sjp90α in mouse liver, we in vitro treated hepatic mononuclear cells derived from *S. japonicum*-infected mice with rSjp40 and rSjp90. FACS analysis indicated significantly increased proportions of macrophages (F4/80^+^ CD11c^−^) (*F*_(3, 16)_ = 4.705, LSD: *p* = 0.0136) (Figure 5a,b), dendritic cells (F4/80^−^ CD11c^+^) (*F*_(3, 16)_ = 6.027, LSD: *p* = 0.0113) (Figure 5a,c), and eosinophilic cells (Siglecf^+^ CD11c^−^) (*F*_(3, 16)_ = 30.25, LSD: *p* < 0.0001) (Figure 5a,d) were detected in these hepatic immune cells after incubation with rSip40. Stimulation with rSjp90α primarily enhanced the response of dendritic cells (F4/80^−^ CD11c^+^) (*F*_(3, 16)_ = 6.027, LSD: *p* = 0.0166) (Figure 5a,c). However, neither rSjp40 or rSjp90α had any effect on granulocytes (Gr1^+^CD11c^−^) (Figure 5e).

CD4^+^ Th cells play essential roles in the initiation and regulation of the hepatic immune response and pathology during schistosome infection [3]. We found in vitro treatment of hepatic mononuclear cells from *S. japonicum*-infected mice with rSjp40 and rSjp90α resulted in an increase in the proportion of CD4^+^ cells (CD90.2^+^CD4^+^) (*F*_(3, 12)_ = 115.0, both LSD: *p* < 0.0001) (Figure 6a,b); these cells orchestrate the development of granulomas in schistosomiasis [3]. In addition, treatment of hepatic mononuclear cells with rSjp40 also induced significantly increased proportions of Th1 cells (CD90.2^+^CD4^+^ IFN-γ^+^) (*F*_(3, 12)_ = 3.551, LSD: *p* = 0.0405), Th2 cells (CD90.2^+^CD4^+^ IL-4^+^) (*F*_(3, 12)_ = 5.226, LSD: *p* = 0.0159), Th17 cells (CD90.2^+^CD4^+^ IL-17^+^) (*F*_(3, 12)_ = 6.623, LSD: *p* = 0.0031), but not Treg cells (CD90.2^+^CD4^+^Foxp3^+^), whereas rSjp90α predominantly induced the expression of Th17 cells (*F*_(3, 12)_ = 6.623, LSD: *p* = 0.0168), but not Th1/2 or Tregs (Figure 6a–f). However, cells treated with SEA, a complex mixture of egg components, and a well-known stimulator of Th2 responses, generated enhanced numbers of CD4^+^ cells (*F*_(3, 12)_ = 115.0, LSD: *p* < 0.0001), Th2 (*F*_(3, 12)_ = 5.226, LSD: *p* = 0.0118), and Treg cells (*F*_(3, 12)_ = 4.373, LSD: *p* = 0.0171) (Figure 6a–f). Of note, we also observed multiple intracellular cytokines in CD90.2^−^CD4^+^ cells (Figure 6e–i), which have been reported to be unusually expressed in the germinal centers of spleen and lymph nodes [33]. The treatment of hepatic mononuclear cells from *S. japonicum*-infected mice with rSjp40 and rSjp90α resulted in an increase in the proportion of CD90.2^−^CD4^+^ cells (*F*_(3, 12)_ = 29.35, both LSD: *p* < 0.0001) (Figure 6g). In addition, the treatment of hepatic mononuclear cells with rSjp40 also significantly stimulated responses of CD90.2^−^CD4^+^IL-4^+^ (*F*_(3, 12)_ = 4.489, LSD: *p* = 0.0159) and CD90.2^−^CD4^+^IL-17^+^ cells (*F*_(3, 12)_ = 4.089, LSD: *p* = 0.0349) (Figure 6h,j), whereas rSjp90α predominantly induced the expression of CD90.2^−^CD4^+^ IFN-γ^+^ (*F*_(3, 12)_ = 3.551, LSD: *p* = 0.0272) and CD90.2^−^CD4^+^ IL-17^+^ cells (*F*_(3, 12)_ = 4.089, LSD: *p* = 0.0223) (Figure 6i,j).

### 2.6. Effect of rSjp40 and rSjp90α on Hepatic Stellate Cells

We investigated the interaction between Sjp40 and Sjp90α with hepatic stellate cells (HSCs), which are the key cellular players in the development of hepatic fibrosis, by stimulating LX-2 cells with rSjp40 and rSjp90α. As noted in Figure 7a,b, we found that the treatment of LX-2 cells with rSjp40 resulted in decreased mRNA levels of classical HSC activation markers such as α-smooth muscle actin (α-SMA) (*F*_(2, 20)_ = 7.918, LSD: *p* = 0.0118) and COL1A1(*F*_(3, 12)_ = 8.438, LSD: *p* = 0.0011), but in contrast, rSjp90α had no significant effect on the expression of these two components.

## 3. Discussion

HSPs, such as HSP40, HSP60, HSP70, and HSP90, are known to be widely expressed by parasites and they play critical roles in the activation and regulation of host innate and antigen-specific adaptive immune responses, the study of which might potentially reveal new anti-parasite vaccine candidates [34,35,36,37]. In the present study, we cloned two HSPs (Sjp40 and Sjp90α) from *S. japonicum*, expressed them in *Escherichia coli* (*E. coli*), and obtained purified soluble recombinant proteins to study their expression and their preliminary characterization. In addition, we report for the first time, an investigation on the roles of Sjp40 and Sjp90α on the regulation of hepatic immune cells. 

Additional features of Sjp40 revealed in this study include its distribution in eggs and adult *S. japonicum* and its effect on mouse hepatic immune cells and hepatic stellate cells following a previous investigation of its diagnostic potential [10]. Sjp90α is found abundantly in the excretory–secretory proteome [7] and in the mRNA [13] in adult *S. japonicum* worms, but no further characterization had been undertaken on this HSP. Accordingly, we isolated Sjp90α from *S. japonicum* and used the recombinant protein for more functional exploits. Our immunolocalization analysis showed Sjp90α was not only expressed throughout the tissues (parenchyma or vitelline cells) of adult worms but was also present in the worm tegument. Given the high homology evident between host HSP90 and Sjp90α, the tegumental location of Sjp90α in adult *S. japonicum* raises concerns whether the signaling we observed by immunolocalization is indeed specifically representing the location of Sjp90α itself rather than host HSp90. Regarding this point, during the immunolocalization procedure, goat serum (1% in blocking buffer) was used to block the worm sections, thereby removing the non-specific background before reacting with primary antibody. In addition, freshly perfused adult worms were washed with perfusion buffer (at least three times) before fixing and the host HSP90 attached to the surface of worms was likely removed. This was reinforced with our observations of negative control worms when normal mouse serum was used as primary antibody (Appendix A), showing host HSP90 was still present in the gut, but there was no signal detectable on the worm surface. We speculate that the location of Sjp90α on the worm surface of *S. japonicum* might help the parasite adapt to the host immune microenvironment and aid it in switching from an immune-sensitive to an immune-tolerance state [38]. Previous work showed killing of *S. japonicum* worms after treatment with the HSP90 inhibitor, geldanamycin, in vitro [39], suggesting Sjp90a is a potential therapeutic target. However, the high identity between Sjp90 and host HSP90 might limit the use of this HSP90 inhibitor.

Both Sjp40 and Sjp90α were found to be present in the Reynolds’ layer within the mature egg, suggesting these proteins are secreted by the miracidium and accumulate between the envelope and the eggshell [31]. We confirmed the expression of Sjp40 and Sjp90α in the egg secretory proteins (ESP), indicating that they are secreted by mature eggs and can interact with surrounding cells. The major immunopathological consequences of *S. japonicum* infection are the egg-induced type 2 immune reaction and the dynamic granulomatous process, which is dependent on the generation of a molecule or molecules attracting eosinophils, neutrophils, macrophages, and other immune cells [40]. Our data show that rSjp40 was able to enhance the expression of macrophages, dendritic cells, and eosinophilic cells; while stimulation with rSjp90α primarily increased the numbers of dendritic cells, which may be induced by the non-homologous domain of Sjp90α, given the high identity between Sjp90α and host HSP90. However, the smaller immune response induced by rSjp90α implies again that this protein might be associated with a species-specific type of immune evasion to assist the schistosome parasite survive in its definitive hosts, a concept supported by a previous study [7].

It has been reported that human HSP40 can bind to the human androgen receptor (AR) [41], a type of progesterone receptors; these have a diverse range of biological actions including important roles in the development and maintenance of the reproductive and immune systems [42]. In addition, there is evidence indicating HSP90 promotes the interaction of steroid hormone with the glucocorticoid receptor (GR), resulting in maintenance hormone binding activity [43]. However, the precise roles that Sjp40 and Sjp90α play in the modulation of host immune responses remain unclear. Indeed, whether Sjp40 interacts with host specific proteins or receptors needs determined in future study. Previous work showed human HSP90α can promote human skin cell migration through CD91 and activate wound healing responses [44]. The high amino acid sequence identity (61%) between Sjp90α and host HSP90 suggests the schistosome HSP may function in granuloma wound healing but this also needs to be further addressed.

It is well recognized that the dominant Th1 response is induced in mice at the early stage (at 3–5 weeks post-infection) of a schistosome infection. With the sexual maturation of adult worms in the host and a large number of eggs released by mature females (at 5–6 weeks post-infection), the host immune response shifts from Th1 to a strong Th2/Th17 response [3]. Our ex-vivo data show that rSjp40 might contribute to the dynamic immune response through inducing the Th1, Th2, and Th17 cellular response, whereas rSjp90α predominantly induced increased Th17 cell response. Previous studies showed that some CD4^+^ T cell populations do not express CD90.2 (CD90.2^−^CD4^+^ cells), which is highly expressed in gut-associated Peyer’s patches [45] and in the germinal centers of spleen and lymph nodes [33]. Our results showed that the unusual CD90.2^−^CD4^+^ cell subsets response appeared to increase in hepatic immune cells after they had been in vitro stimulated by either rSjp40 or rSjp90α, suggesting that the growth of these cell subsets may be associated with the process of immune regulation during *S. japonicum* infection. In addition, we found that the expression of fibrogenic gene markers (α-SMA and COL1A1) was decreased in human HSC after being incubated with rSjp40 but not with rSjp90α. This suggests Sjp40 might play a pivotal role in the suppression of liver fibrosis, whereas Sjp90α appears to play no part in this process.

## 4. Methods

### 4.1. Ethics Statement

The conduct and procedures involving animal experimentation were approved by the Animal Ethics Committee of QIMR Berghofer Medical Research Institute (project number 288 and ethics ID A0108-054). This study was performed in accordance with the recommendations in the Guide for the Care and Use of Laboratory Animals of the USA National Institutes of Health.

### 4.2. Parasites

*S. japonicum* adult worms were collected by perfusion of ARC Swiss mice infected percutaneously with 60 cercariae of *S. japonicum* (Anhui population, mainland China), shed from *Oncomelania hupensis* snails, which were transported to the QIMR Berghofer Medical Research Institute in Australia. Female Swiss mice were percutaneously infected with *S. japonicum* cercariae. Adult worms were collected from infected mice 6 weeks post infection. Mice were perfused with perfusion buffer [46] to remove worms from mesenteric veins. Eggs were harvested from mouse liver as described [47]. Briefly, livers were minced with scissors and digested in PBS containing 100 mg/mL Collagenase B (Roche, Mannheim, Germany) and 1% (*v/v*) penicillin/streptomycin (Gibco, Waltham, MA, USA) overnight at 37 ℃. The digested eggs were then centrifuged at 500× *g* for 5 min, followed by washing with ice-cold PBS 3 times. Then the eggs were collected by passing through 250 μm and 150 μm cell strainers, separately, and then purified using a Percoll (GE Healthcare, Buckinghamshire, UK) gradient.

### 4.3. Bioinformatic Analysis

BLAST of Sjp40 (Sjp_0021240) and Sjp90α (Sjp_0044660) was performed in the Wormbase (https://parasite.wormbase.org). The BioEdit software was used for multiple sequence alignments. The motif domains of Sjp40 and Sjp90α were analyzed using the ScanProsite of ExPASy analysis tools (https://prosite.expasy.org). Sjp40 and Sjp90α homologous sequences were analyzed by the neighbor-joining method for phylogenetic analysis and the tree was plotted using MEGA-X software [48]. The demonstration of the Sjp40 and Sjp90α tertiary structure was created by online SWISS-MODEL (https://swissmodel.expasy.org) [49]. ATP-binding residues for Sjp90 were predicted by NsitePred (http://biomine.cs.vcu.edu/servers/NsitePred/) [50].

### 4.4. Preparation of Soluble Egg Antigen and ESP

Soluble egg antigen (SEA) of *S. japonicum* were prepared as described [47] with some modification. Briefly, eggs were suspended in 1 × PBS plus protease inhibitor cocktail (Sigma-Aldrich, Milwaukee, WI, USA) and homogenized using a Precellys Homogeniser (Stretton Scientific, Derbyshire, UK) at 5500 rpm for 30 s at 4 ℃. Following centrifugation at 16,000× *g* for 1 h, the supernatant (SEA) was collected. Residual endotoxin in the SEA extracts was determined using a Pierce LAL Chromogenic Endotoxin Quantitation Kit (Thermo Scientific, Rockford, USA) according to the manufacturer’s instructions.

For the preparation of excretory secretory protein (ESP) antigens from eggs, approximately 30,000 eggs/well were cultured in a 24-well plate with 1 mL RPMI1640 and 1% (*v/v*) penicillin/streptomycin at 37 °C. Culture supernatant was collected twice with a three-day interval and replaced by 1 mL fresh media. The mixture of supernatants was then filtered through a 20 µm strainer and concentrated using a 3–5 KDa MW filter.

### 4.5. Escherichia Coli Protein Expression

A 1065 bp fragment of cDNA encoding Sjp40 and a 780 bp fragment of cDNA encoding Sjp90α were amplified by PCR and inserted into the pET28b vector (Invitrogen, Carlsbad, CA, USA). *E. coli* BL21 (DE3) cells (Invitrogen) were transformed with the reconstructed plasmids for protein expression as described [51]. The expressed rSjp40 and rSjp90α proteins were purified from *E. coli* lysates using Ni-NTA affinity chromatography (GE Health Life Science, Pittsburgh, PA, USA). Given previous studies showing the 6 × His tag did not stimulate any immunogenic activity [52,53,54,55], the tag was not removed after purification of rSjp40 and rSjp90α. However residual endotoxin was removed from both recombinant proteins as described [51] and Endotoxin (*E. coli*) Standards kits (Lonza, Basel, Switzerland) were used subsequently to confirm there was no Endotoxin contamination in either protein.

### 4.6. Preparation of Sjp40 and Sjp90α Antibody

Sjp40 and Sjp90α antiserum were prepared in Swiss mice. Recombinant Sjp40 or Sjp90α were formulated with either Complete Freund’s Adjuvant (Sigma-Aldrich, Castle Hill, Australia) (primary injection) or Incomplete Freund’s Adjuvant (two boosts at two weekly intervals) (Sigma-Aldrich) and the preparations were subcutaneously injected into the mice. Mouse bloods were collected 2 weeks after the last injection using cardiac puncture and sera obtained. Antibody titres were determined using an enzyme-linked immunosorbent assay (ELISA) against rSjp40 (1:128,000) or rSjp90α (1:128,000) as described [56].

#### 4.6.1. Western Blotting

For Western blot analysis, equivalent amounts of the Sjp40 and Sjp90α recombinant proteins (~10 μg) or protein (~200 μg) from SEA or ESP were loaded into each lane for separation by SDS-PAGE and electrophoresed proteins were then transferred onto Immu-Blot^®^ PVDF membranes (Bio-Rad, Hercules, CA, USA). After blocking in Odyssey Blocking Buffer for 1 h at room temperature, membranes were incubated at 4 °C overnight with primary antibodies: mouse anti-rSjp40 or mouse anti-rSjp90α, respectively. After washing with PBS plus 0.1% Tween 20 with gentle shaking, the membranes were incubated at room temperature for 1 h with IRDye^®^ secondary antibody (1:20,000 dilution) in Odyssey Blocking Buffer plus 0.1–0.2% Tween 20. After washing, the blots were developed using the Odyssey Imaging System.

#### 4.6.2. Immunohistochemistry

Liver tissue from mice infected with *S. japonicum* or adult worms perfused from infected mice with perfusion buffer (0.85% (*w/v*) NaCl plus 1% (*w/v*) trisodium citrate) were fixed in 10% formalin, embedded in paraffin and sectioned at 3 µm. Following antigen retrieval by boiling in 0.01 M sodium citrate, pH 6, for 20 min in a water bath, endogenous peroxidase activity was blocked by incubation with 3% (*v/v*) H_2_O_2_ for 20 min at room temperature. Slides were washed three times with PBS and blocked with 5% bovine serum albumin and 1% (*w/v*) goat serum in tris buffered saline (TBS) for 30 min at room temperature. Tissue sections were then probed with 1:100 diluted mouse anti-rSjp40 or rSjp90α sera for 1 h at 37 °C (normal mouse serum was used as negative control). Slides were then washed three times with 1 × PBS, and HRP conjugated secondary antibody (Sigma) was added as secondary antibody at a dilution of 1:1000 and incubated for 10 min at room temperature. Sections were examined with an Aperio Scanscope XT scanner (Aperio Technologies, Vista, CA, USA) and analyzed using ImageScope software (Aperio, ImageScope version 10.2.1.2315).

To confirm the integrity of the HRP labelling of Sjp40 and Sjp90α, we replaced the HRP conjugated secondary antibody with Alexa-Fluor 647 conjugated antibody for the immunolocalisation analysis. Briefly, the fixed liver tissue or adult worm sections were treated with Revealt A solution (Biocare Medical, Concord, MA, USA) at 120 °C for 5 min for antigen retrieval. After the non-specific binding sites were blocked with 1% (*w/v*) bovine serum albumin and 1% (*w/v*) goat serum in TBS for 60 min at 4 °C, the liver tissue or worm sections were stained with mouse anti-mouse Sjp40 or Sjp90α overnight at 4 °C as described [27] (normal mouse sera were used as negative control). After staining, excess antibody was removed by four washing steps with TBST (0.05% (*v/v*) Tween 20 in TBS). Then the liver tissue or sections were incubated with Alexa-Fluor 647 goat anti-mouse IgG (1:500) (Invitrogen, Carlsbad, CA, USA) at 37 °C for 1 h. Diamidino-2-phenylindole (DAPI) gold (Invitrogen) was used to stain for nuclei and the samples were analyzed using a Zeiss 780 NLO laser (Zeiss, Oberkochen, Germany).

#### 4.6.3. Quantitative RT-PCR (qRT-PCR)

Total RNA was extracted from human hepatic stellate or different life cycle stages of *S. mansoni* (including female worms, male worms, schistosomula, eggs and cercariae; miracidia was not included due to limited materials) using RNeasy Mini Kits (Qiagen, Hilden, Germany). First strand cDNA was synthesized using a Sensiscript Reverse Transcription for First strand cDNA synthesis Kit (Qiagen) and was subsequently used as template in qPCR to determine the expression levels of Sjp40 and Sjp90α in different life cycle stages and the α-SMA and COL1A1 expression levels in hepatic stellate cells. *S. japonicum* PSMD4 (26S proteasome non-ATPase regulatory subunit 4) [27] or the human housekeeping gene β-actin [57] was used as the reference gene. Real-time PCR was performed using Power Syber Green PCR Master Mix (Applied Biosystems, Foster City, CA, USA) and data were processed using Rotor Gene 6000 series software (Corbett Life Science, Australia). The cycling parameters were as follows: 95 °C for 3 min; 30 cycles of 95 °C for 30 s and 54 °C for 30 s, and 70 °C for 5 min. Relative expression levels were calculated using the comparative CT method (2^−ΔΔCt^) [58]. Then, mRNA levels of each *S. japonicum* stage were normalized to the data obtained with female worms, based on their suitably expressed CT values. The sequences of primers used in this analysis are shown in Appendix A.

### 4.7. Preparation of Liver Non-Parenchymal Cells (NPC)

Liver nonparenchymal cells (NPCs) were isolated following an established method with some modifications [59]. Briefly, five livers collected from mice at 6-week post *S. japonicum* infection were cut into small pieces and digested in 20 mg Collagenase B at 37 ℃ for 45 min. The digested liver tissue was then mashed through a 100 µm filter, followed by centrifugation at 400 g for 6 min. The cell pellets were purified using Percoll gradients (GE Healthcare) by centrifuging at 690× *g* for 12 min at room temperature. After discarding the upper layers carefully, the cell pellets were collected and resuspended in red cell lysis buffer (Sigma-Aldrich) and then washed in 10 mL complete RPMI1640 (Gibco, Grand Island, NY, USA) containing 10% (*v/v*) FBS (Gibco) and 1% (*v/v*) penicillin/streptomycin (Gibco). Then, the single cell suspension was collected through the Pre-Separation Filters (20 μm, Miltenyi Biotec, Bergisch Gladbach, Germany) for further study.

#### 4.7.1. Flow Cytometry

For Th1/Th2/Th17 cells analysis, 2 × 10^6^ of single cell suspensions were cultured in complete RPMI 1640 medium and stimulated with rSjp40, rSjp90α, or SEA, at 20 μg/mL for 44 h; then the cells were mixed with 25 ng/mL PMA (Sigma) and 1 μg/mL ionomycin (Sigma) in the presence of 0.66 μL/mL Golgistop (BD Bioscience, Melbourne, Australia) at 37 °C in 5% CO_2_ for 4 h. Cells were blocked with Fc receptor (BD Bioscience) and stained with the Live/Dead-Horizon stain (Thermo Fisher Scientific, 1:1000 dilution) for 15 min, surface-stained with CD90.2 (Thy1.2)-PEcy7 and CD4-BV785 antibodies (BD Bioscience, 1:200 dilution) for 30 min, then cells were fixed and permeabilized with BD Cytofix/Cytoperm buffer (BD Bioscience) as per the manufacturer’s protocol and intracellularly stained with IFN-γ-Percp/cy5.5, IL-4-PE-CF594, and IL-17A-AF647 (BD Bioscience, all 1:100 dilution) for 30 min.

For Treg cells analysis, 2 × 10^6^ of single cell suspensions were incubated with rSjp40 (20 μg/mL), rSjp90α (20 μg/mL), or SEA (20 μg/mL) for 48 h, followed by staining with the Live/Dead-Horizon stain (Thermo Fisher Scientific, 1:1000 dilution) for 15 min, and then surface-stained with CD90.2 (Thy-1.2)-PEcy7 and CD4-BV785 (BD Bioscience, 1:200 dilution) for 30 min. Then, cells were then fixed and permeabilized with fixation/permeabilization buffers (BD Bioscience) for 50 min and then blocked with Fc receptor (BD Bioscience). Finally, cells were stained with PE-conjugated anti-Foxp3 antibodies (BD Bioscience, 1:200 dilution) for 30 min.

For other hepatic immune cells analysis, 2 × 10^6^ of single cell suspensions were stimulated with rSjp40, rSjp90α, or SEA, at 20 μg/mL for 48 h and stained with the Live/Dead-Horizon stain (Thermo Fisher Scientific) for 15 min and then surface-stained with Gr-1-AF700, CD11c-APC/cy7, F4/80-BV421, and Siglecf-BV605 (all from BD Bioscience) for 30 min. All the stained cells were collected using Fortessa 4A and FACS Diva software (BD Bioscience). Flow cytometry analysis was performed using FlowJo version 10 software (TreeStar, Ashland, OR, USA).

#### 4.7.2. LX-2 Human Hepatic Stellate Cell Culture and Treatment

A total of 5 × 10^5^ of LX-2 cells/well were cultured in six-well plates containing DMEM supplemented with 10% (*v/v*) FBS and 1% (*v/v*) penicillin/streptomycin, at 37 ℃ in a humidified atmosphere containing 5% CO_2_. LX-2 cells were stimulated with PBS, rSjp40 (20 μg/mL) or rSjp90 (20 μg/mL) for 48 h, respectively; then the cells were collected by centrifugation at 500 g for 5 min. Total RNA was extracted from these cells and used as template to synthesize cDNA which was used for RT-PCR analysis to determine the expression level of fibrosis-related genes (α-SMA and COL1A1) as described above.

### 4.8. Statistical Analysis

All analyses were carried out with GraphPad software (Version 7.02). Data are shown as means ± SEM. The significance of the difference between two groups was determined using Student’s t test. Multiple comparisons were performed by one-way ANOVA, followed by LSD post-testing for comparisons between two groups. *F* values and *p* values were calculated by GraphPad software. *p* values < 0.05 were considered significant. The *F*_(dfn, dfd)_ distribution has two parameters: Degrees of freedom numerator (dfn) and degree of freedom denominator (dfd).

Dfn = the number of groups−1; dfd = the total number of subjects in the experiment-dfn.

## 5. Conclusions

To better understand the roles of HSPs in schistosomes, we isolated and characterized two HSPs, Sjp40 and Sjp90α, from *S. japonicum*. Immunolocalisation of these proteins in eggs and in adult *S. japonicum* and their contribution in the development of hepatic immunopathology, potentially provides new insights into the disease of schistosomiasis. In addition, in contrast to Sjp40, Sjp90α being located on the tegument of adult *S. japonicum* may play an important role in helping the parasite adapt to the host immune microenvironment, as it switches from an immune-sensitive to an immune-tolerance state.

## Figures and Tables

**Figure 1 ijms-21-04034-f001:**
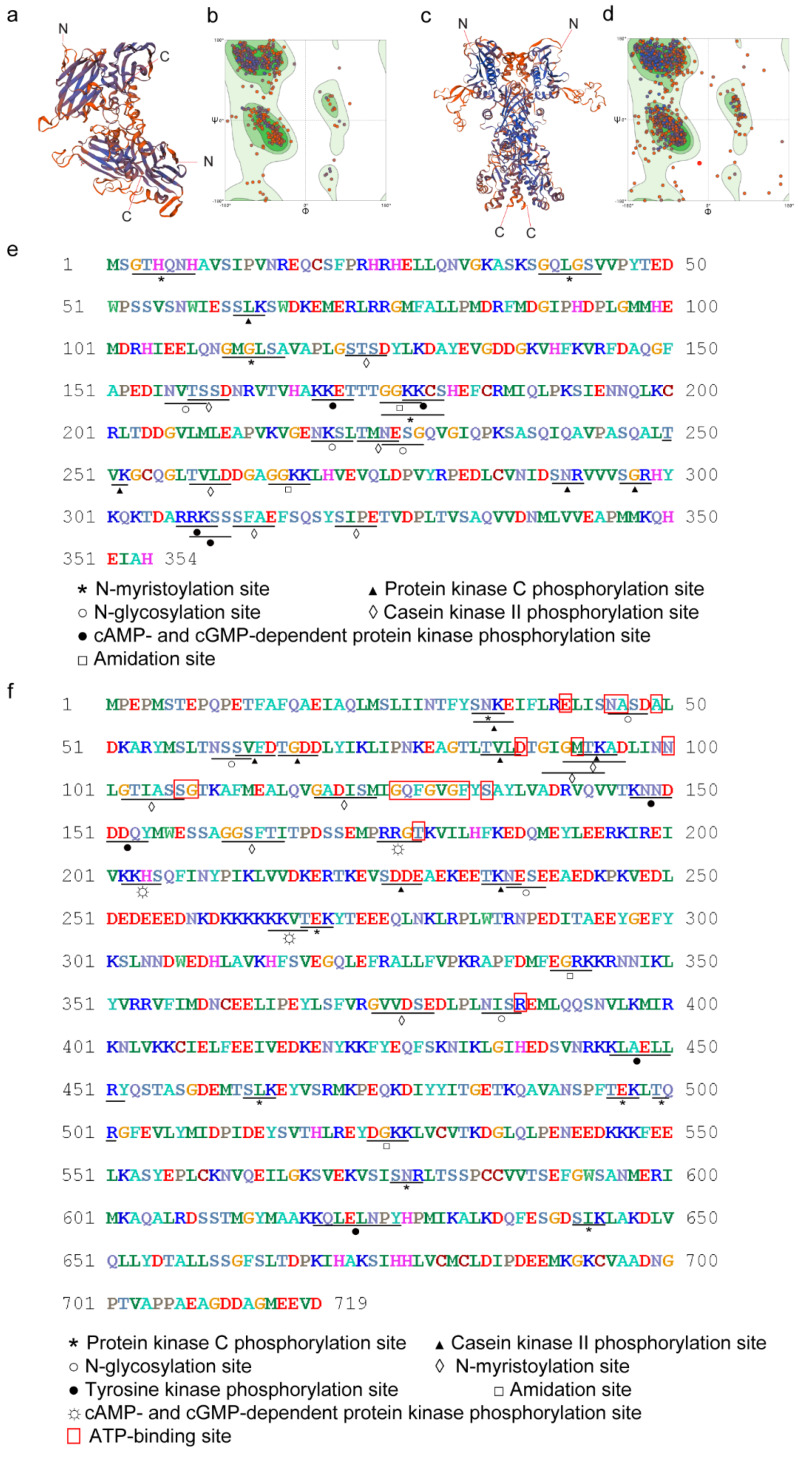
Predicted 3D structure and motifs of Sjp40 and Sjp90α. Homology modeling was performed using the Swiss-Model Server and the predicted 3D structure of homo-dimer Sjp40 (**a**) and homo-dimer Sjp90 (**c**) are shown. Model quality was evaluated using the Ramachandran plot method and the results represent the acceptable stability of the 3D structure of Sjp40 (**b**) and Sjp90 (**d**). (**e**) Sjp40 motifs: *: N-myristoylation site; ▲: protein kinase C phosphorylation site; ◊: casein kinase II phosphorylation site; ○: N-glycosylation site; ●: cAMP- and cGMP-dependent protein kinase phosphorylation site; and □: amidation site. (**f**) Sjp90 motifs: *: protein kinase C phosphorylation site; ▲: casein kinase II phosphorylation site; ◊: N-myristoylation site; ●: tyrosine kinase phosphorylation site; □: amidation site; ☼: cAMP- and cGMP-dependent protein kinase phosphorylation site; and □: ATP-binding site.

**Figure 2 ijms-21-04034-f002:**
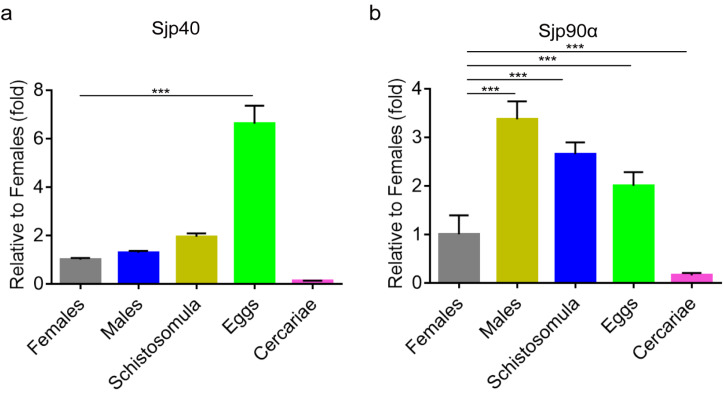
Expression levels of Sjp40 and Sjp90α in different life cycle developmental stages of *S. japonicum*. RT-PCR analysis was performed to measure the transcription of Sjp40 (**a**) and Sjp90α (**b**) in female and male adult worms, schistosomula, eggs and cercariae (*S. japonicum* PSMD4 was used as house-keeping gene); then the transcription levels of Sjp40 and Sjp90α were normalized to those of female worms, which had certain expressed CT values for comparison. Data are represented as the mean of two independent experiments with SEM. Multiple comparisons were performed by one-way ANOVA, followed by Least Significant Difference (LSD) post hoc test for comparisons between two groups *** *p* < 0.001 significant differences compared with the adult female stage.

**Figure 3 ijms-21-04034-f003:**
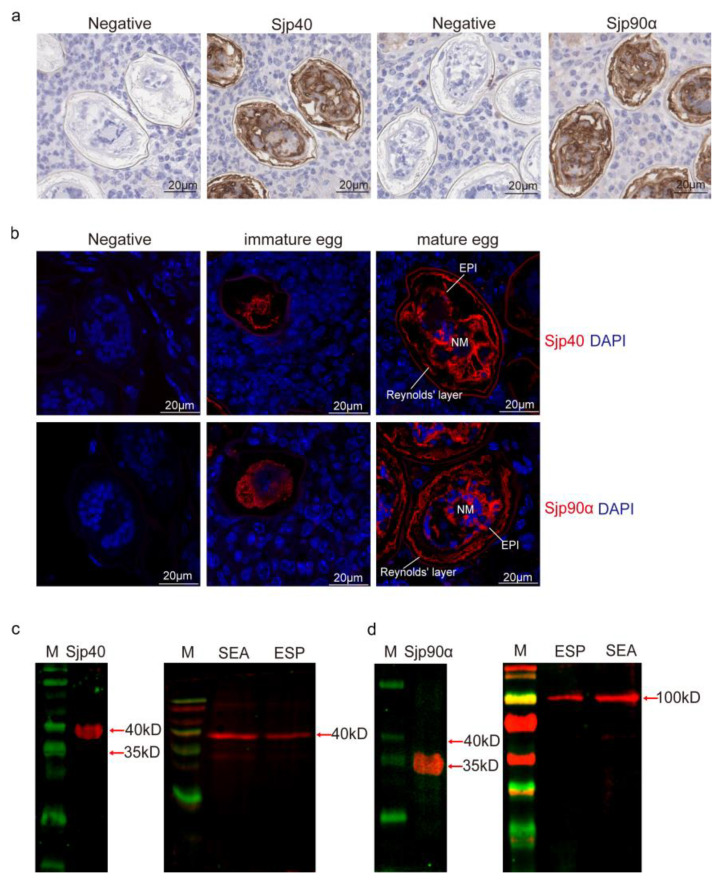
Native Sjp40 and Sjp90α identified in eggs and egg secreted proteins (ESP) of *S. japonicum.* Immunolocalization of Sjp40 and Sjp90α in eggs trapped in infected mouse liver probed with mouse anti-rSjp40 and anti-rSjp90 antibodies, respectively. Sections of livers from mice infected with *S. japonicum* were incubated with naïve control mouse serum (negative control) and mouse anti-rSjp40 or anti-rSjp90α antiserum, and subsequently with horseradish peroxidase (HRP)-conjugated secondary antibody (**a**) or Alexa-Fluor 647-conjugated secondary antibody (**b**). DAPI stained nuclei are blue. A section of schistosome eggs in a developing granuloma shows SjP40 or Sjp90 staining (brown in HRP labelling and red in Alexa-Fluor 647) in eggs. NM—neural mass; EPI—epidermal cells. Scale-bars: 20 μm. (**c**,**d**) Western blotting was used to detect recombinant Sjp40 (**c**, left lane) and Sjp90α (**d**, left lane). Anti-rSjp40 (**c**, right lane) and anti-rSjp90 (**d**, right lane) were used to probe protein extracts of the soluble egg antigen (SEA) and egg secreted protein (ESP).

**Figure 4 ijms-21-04034-f004:**
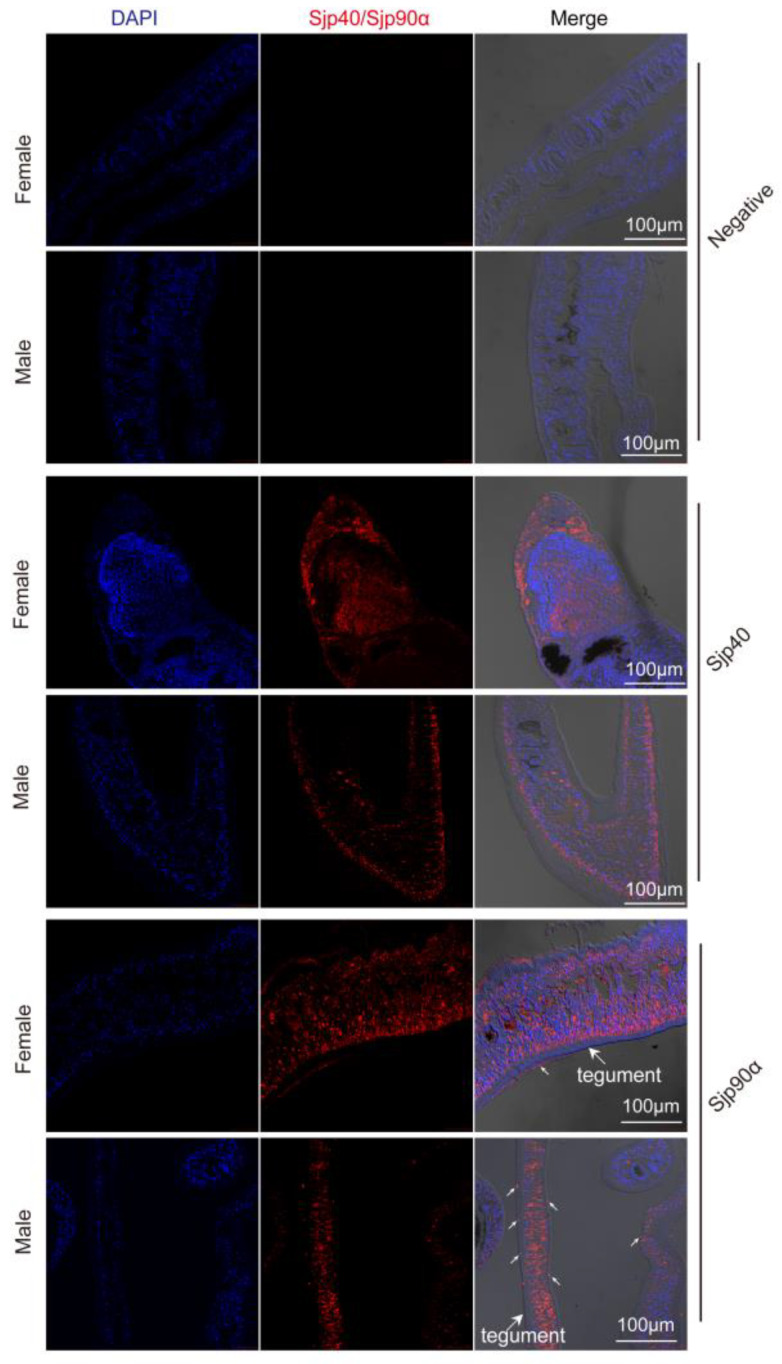
Distribution of Sjp40 and Sjp90α in adult *S. japonicum*. Immunolocalization of Sjp40 and Sjp90α in the worms of *S. japonicum*. Adult male and female worm sections were labelled with mouse anti-rSjp40 and anti-rSjp90α antibody coupled with Alexa-Fluor 647 goat anti-mouse IgG (red), DAPI was used to stain for nuclei and the samples were analyzed using a Zeiss 780 NLO laser. Negative control sections of the worm were incubated with normal mouse serum. Scale-bars: 100 μm. White arrows indicate the Sjp90α expressed in the tegument of worms.

**Figure 5 ijms-21-04034-f005:**
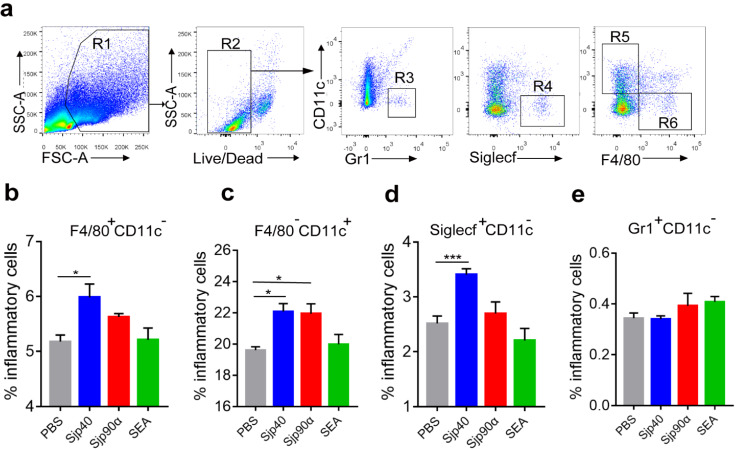
Sjp40 and Sjp90α induce hepatic inflammatory responses. Hepatic immune cells (2 × 10^6^) from *S. japonicum*-infected mice were stimulated with rSjp40 (20 μg/mL), rSjp90α (20 μg/mL) and SEA (20 μg/mL) for 48 h, respectively, then the proportions of hepatic immune cell subsets were investigated. Gating strategy (**a**) and the percentages of macrophages (F4/80^+^ CD11c^−^) (**b**), dendric cells (F4/80^−^ CD11c^+^) (**c**), eosinophilic cells (Siglecf^+^ CD11c^−^) (**d**), and granulocytes (Gr1^+^CD11c^−^) (**e**) in inflammatory cells were analyzed by FACS. Data are expressed as the mean ± SEM (*n* = 5 each group), *** *p* < 0.001, * *p* < 0.05 (ANOVA/LSD).

**Figure 6 ijms-21-04034-f006:**
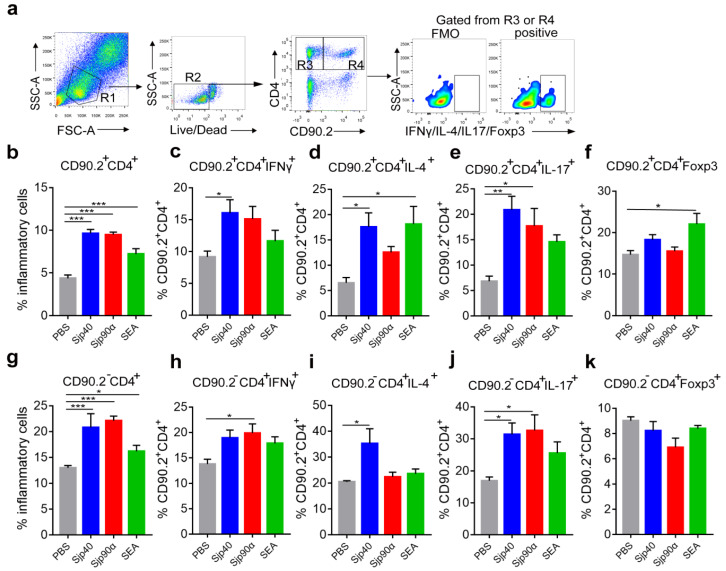
Mouse liver CD4^+^ T cells response induced by rSjp40 and rSjp90α. Hepatic immune cells (2 × 10^6^) from *S. japonicum*-infected mice were treated with rSjp40 (20 μg/mL), rSjp90α (20 μg/mL) and SEA (20 μg/mL) following PMA/ionomycin stimulation. The gating strategy (**a**) and proportion of CD4^+^ cells (CD90.2^+^CD4^+^) (**b**), Th1 cells (CD90.2^+^CD4^+^IFN-γ^+^) (**c**), Th2 cells (CD90.2^+^CD4^+^IL-4^+^) (**d**), Th17 cells (CD90.2^+^CD4^+^IL-17^+^) (**e**), Treg cells (CD90.2^+^CD4^+^Foxp3^+^) (**f**), CD90.2^−^CD4^+^ (**g**), CD90.2^−^CD4^+^IFN-γ^+^ (**h**), CD90.2^−^CD4^+^IL-4^+^ (**i**), CD90.2^−^CD4^+^IL-17^+^ (**j**), and CD90.2^−^CD4^+^Foxp3^+^ cells (**k**) in lymphocytes were analyzed by FACS. Data are shown as the mean ± SEM (*n* = 4 each group), *** *p* < 0.001, ** *p* < 0.01, * *p* < 0.05 (ANOVA/LSD).

**Figure 7 ijms-21-04034-f007:**
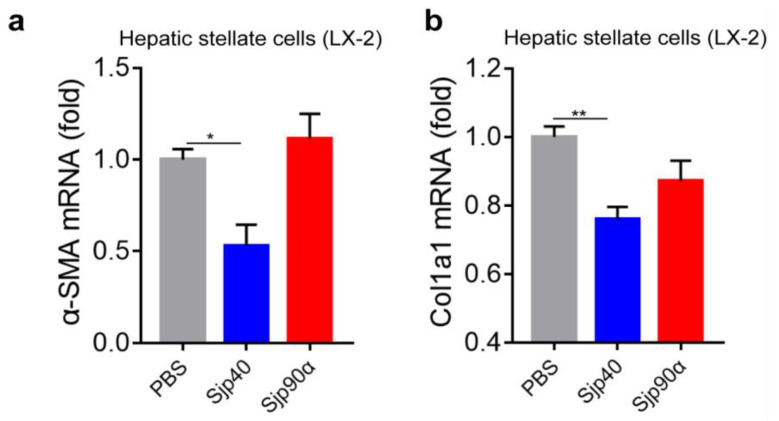
Different effects of rSjp40 and rSjp90α on α-SMA and COL1A1 expression in LX-2 cells. Transcript expression levels of α-SMA (**a**) and COL1A1 (**b**) in LX-2 cells co-cultured with PBS, rSjp40 and rSjp90α, respectively. β-actin was used as the internal standard for normalization. Data are expressed as the mean ± SEM (*n* = 8 each group), ** *p* < 0.01, * *p* < 0.05, compared with PBS control.

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
