# Peer review of "A Biological and Immunological Characterization of Schistosoma Japonicum Heat Shock Proteins 40 and 90α"

_ijms, 2020, doi:10.3390/ijms21114034_

Round 1

Reviewer 1 Report

This manuscript needs an intense revision due to the inacceptable presentation of figure 1,  in principle the quality of all figures have a too low resolution and must be improved.

The modeling result obtained by (swiss model) is not a convincing result, the modeling parameters must be given in a state of the art. The position of ATP binding site middle domain and C-terminal domain must be indicated clearly for Hsp90, was shown the dimer or monomer? The alignment to the selected crystal structure should be indicated in the model and more the ATP position in the pocket should be shown. Also the resolution of the supplementary picture too low and the letters are not visible. But what is the information in Fig 1, conserved binding sites are different to other orthologues? if this occurs, ATPase activity assay and/or susceptibility for 17AAG should be included.

Another critical point is whether the tag was removed or not that means if not, all experiments with purified protein and measurements of the inflammatory reaction must be repeated.

Some information should be given about Th cells.

Due to the interesting thematic impact I recommend acceptance after intense revision of the manuscript

Author Response

Dear reviewers,

We appreciate so much the helpful and constructive comments from both reviewers. We have extensively revised our manuscript according to the reviewers’ suggestions. The newly revised manuscript further supports the conclusions described in the manuscript.

Our point-by-point responses to the reviewers’ comments are described below. We now hope the MS is now suitable for publication.

Reviewer 1

  1. This manuscript needs an intense revision due to the inacceptable presentation of figure 1, in principle the quality of all figures have a too low resolution and must be improved.

Response: We apologize for the poor quality of the Figs. We have now uploaded all figures with high resolution (including the Supplementary Figs).

  1. The modeling result obtained by (swiss model) is not a convincing result, the modeling parameters must be given in a state of the art.

Response: As suggested, we have added the swiss modeling parameters in the revised MS on lines 85-93.

  1. The position of ATP binding site middle domain and C-terminal domain must be indicated clearly for Hsp90, was shown the dimer or monomer? The alignment to the selected crystal structure should be indicated in the model and more the ATP position in the pocket should be shown.

Also the resolution of the supplementary picture too low and the letters are not visible. But what is the information in Fig 1, conserved binding sites are different to other orthologues? if this occurs, ATPase activity assay and/or susceptibility for 17AAG should be included.

Response: We thanks the reviewer for these comments. Fig 1 has now been modified showing the ATP binding sites. As required, we re-searched the ATP binding sites online NsitePred (http://biomine.cs.vcu.edu/servers/NsitePred/) by using full length sequences of Hsp90 and found 19 ATP-binding sites located at N-terminal of SjP90α. All these ATP-binding sites are extremely conserved as typified by the murine heat shock protein HSP90 (NP_032328.2) (E41xxxN45A46xxxA49; D87xxxxM92; N100xxxxxxS107G108; GQFGVGF(162-132)xS134; T178; R387). We have added this information on lines 125-135.

  1. Another critical point is whether the tag was removed or not that means if not, all experiments with purified protein and measurements of the inflammatory reaction must be repeated.

Response: We did not remove the HIS tag, given previous studies showing the 6xHis tag with a small molecular weight (1.2KD) did not present any immunogenic activity (see references 1-3 below).

  1. Some information should be given about Th cells.

Response: We have now added information about Th cells in the revised MS on line 25 and lines 247-248.

References

  1. Zhao, X.; Li, G.; Liang, S., Several affinity tags commonly used in chromatographic purification. Journal of analytical methods in chemistry 2013, 2013, 581093;
  2. Loughran, S. T.; Walls, D., Purification of poly-histidine-tagged proteins. Methods Mol Biol 2011, 681, 311-35;
  3. Ghose, C.; Eugenis, I.; Sun, X.; Edwards, A. N.; McBride, S. M.; Pride, D. T.; Kelly, C. P.; Ho, D. D., Immunogenicity and protective efficacy of recombinant Clostridium difficile flagellar protein FliC. Emerging microbes & infections 2016, 5, e8.

Reviewer 2 Report

In this study, Dr. Xu and colleagues have characterized both biologically (protein sequences homology and modelisation of the tertiary structures, stage-specific expressions and immunolocalizations) and immunologically (impact of these proteins on hepatic immune cells and stellate cells) two heat shock proteins (HSP40 and 90 alpha) of the human blood fluke Schistosoma japonicum. Knowing better the overall characteristics of this two protein is particularly important has heat shock proteins have been shown to be secreted by schistosome parasites and are able to trigger host immune response. Overall, the study is well designed, the methodology is robust enough and results are relatively clearly exposed and are convincing. The manuscript is well written and easy to read. However, I have some questions/suggestions (see below) that should be addressed by the authors before the full acceptance of the present manuscript.

Decision: Accepted with minor revisions

======= LEGENDS =======
p.: page
§: paragraph
l.: line
Fig.: figure
Tab.: table
> : replaced by

======= MAJOR COMMENTS =======
------- COMMENTS ON THE TEXT -------
- Major comments -
Results:
l. 82: The authors did not provide the % of sequence identity between S. japonicum and S. haematobium for HSP90 but they did for HSP40. Why? This information should be stated in the manuscript.
Looking at the predicted tertiary structures of the two HSP studied, do the authors have plan to crystalize these proteins and compare the crystal structure to the predicted 3D structure provide by the Swiss-Model tool? I totally understand that is not an easy experiment to do but, at some point, might be very interesting to validate the modelized 3D structures of these two proteins.

l.119: Stage specific mRNA expression of Sjp40 and Sjp90 alpha - It would have been great to use at least 2 housekeeping genes to normalize the RT-qPCR data.
The authors have tested the expression of Sjp40 and Sjp90 alpha in eggs, cercariae, schistosomula and adult worms (male and female). Why the authors do not tested these genes expression in free miracidia? This should be at least explained in the manuscript (Methods sections).
On the same manner, the authors did not explain why they chose to normalization their data using to expression in female worms. This should be explain somewhere in the method section. Moreover, statistical analysis should also appear in the text (name of the test used, value, df, p-value), as it is not very clear for the reader which comparisons are statistically significant.

l.162: Immunolocalization of Sjp40 and Sjp90 alpha in adult worms - The authors have measured the level of antibody in mice sera for anti-Sjp90 alpha but never mentioned the level of antibody in mice sera for anti-Sj40. Why? Do the authors tested this levels of antibody for anti-Sjp40: if yes, this should be mentioned in the manuscript and the authors should provide some supplementary data as they did for anti-Sjp90 and if no, the authors have to explain why this wasn't tested.

Figures:
- Figure 2: On the graph, no stars (corresponding to significant p values indicated in the figure legend) appear but also any mention of non-statistically significant comparisons. Is it a mistake or no results are significantly different? This should be corrected.

Material and Methods:
l. 392: Quantitative RT-qPCR - The 2 delta delta Ct method is not the most accurate to analyze RT-qPCR data. The authors should have used the efficiency delta Ct method. For that, the authors should have first determine the efficiency of their respective pairs of primers using standard curves and use these efficiency values to calculate way more accurately relative gene expression (see Rao X, Huang X, Zhou Z, Lin X. An improvement of the 2ˆ(-delta delta CT) method for quantitative real-time polymerase chain reaction data analysis. Biostat Bioinforma Biomath. 2013 Aug;3(3):71-85. PMID: 25558171; PMCID: PMC4280562.)

======= MINOR COMMENTS =======

------- COMMENTS ON THE TEXT -------
- Minor comments -
Introduction:
l. 69: "hos", please replace by "host" (t is missing).

----- COMMENTS ON THE REFERENCES -----
None

Author Response

Dear reviewers,

We appreciate so much the helpful and constructive comments from both reviewers. We have extensively revised our manuscript according to the reviewers’ suggestions. The newly revised manuscript further supports the conclusions described in the manuscript.

Our point-by-point responses to the reviewers’ comments are described below. We now hope the MS is now suitable for publication.

Reviewer 2

- Major comments -

Results:

  1. 82: The authors did not provide the % of sequence identity between S. japonicum and S. haematobium for HSP90 but they did for HSP40. Why? This information should be stated in the manuscript.

Response: We apologize for the missing information. As suggested, we have added % of sequence identity between Hsp90 for S. japonicum and S. haematobium on lines 82~83.

  1. Looking at the predicted tertiary structures of the two HSP studied, do the authors have plan to crystalize these proteins and compare the crystal structure to the predicted 3D structure provide by the Swiss-Model tool? I totally understand that is not an easy experiment to do but, at some point, might be very interesting to validate the modelized 3D structures of these two proteins.

Response: We appreciate the reviewer’s suggestion; it will indeed be very interesting and informative to crystalize the HSPs and compare the crystal structure with the predicted 3D structure. This is an important area we hope to advance in the future.

  1. 119: Stage specific mRNA expression of Sjp40 and Sjp90 alpha - It would have been great to use at least 2 housekeeping genes to normalize the RT-qPCR data.

Response: In many previous publications, PSMD4 (26S proteasome non-ATPase regulatory subunit 4, GenBank accession number: FN320595) has been validated as a standard alone reference gene in transcriptomic analysis of S. japonicum4, and has been widely used as reference gene for transcriptomic analysis of S. japonicum4-6.

  1. The authors have tested the expression of Sjp40 and Sjp90 alpha in eggs, cercariae, schistosomula and adult worms (male and female). Why the authors do not tested these genes expression in free miracidia? This should be at least explained in the manuscript (Methods sections). On the same manner, the authors did not explain why they chose to normalization their data using to expression in female worms. This should be explain somewhere in the method section. Moreover, statistical analysis should also appear in the text (name of the test used, value, df, p-value), as it is not very clear for the reader which comparisons are statistically significant.

Response: We did detect transcription of Sjp40 and Sjp90α in miracidia but unstable and very low expression levels were observed for this stage. We comment on this in the revised MS

The transcription levels of Sjp40 and Sjp90α were normalized to female worms, which had certain expressed CT values for comparison. We have now modified the legend of Fig. 2 in the revised MS (lines 184-185).

The statistical analysis information is added in the text and the figure legends. The statistical methods have also been provided in the methods section 4.8 statistical analysis (line 523-527).

  1. 162: Immunolocalization of Sjp40 and Sjp90 alpha in adult worms - The authors have measured the level of antibody in mice sera for anti-Sjp90 alpha but never mentioned the level of antibody in mice sera for anti-Sj40. Why? Do the authors tested this levels of antibody for anti-Sjp40: if yes, this should be mentioned in the manuscript and the authors should provide some supplementary data as they did for anti-Sjp90 and if no, the authors have to explain why this wasn't tested.

Response: Previous study7 demonstrated Sjp40 was a potential diagnostic antigen for early stage parasite detection. Sjp40 IgG antibodies were detected on day 21 post infection in mice infected with 50 cercariae7, but this represents a very heavy infection. In the current paper our ELISA experiments aimed to determine whether Sj90α has potential as a diagnostic marker of early infection using a low dosage of cercariae infection. We found anti-Sjp90α was undetectable until week 7 after mice had been infected with 12 cercariae. Consequently as this result was not particularly encouraging, we decided to omit this part.

6. Figures: Figure 2: On the graph, no stars (corresponding to significant   p values indicated in the figure legend) appear but also any mention of non-statistically significant comparisons. Is it a mistake or no results are significantly different? This should be corrected.

Response: The statistical analysis results have now been added as required.

  1. Material and Methods:392: Quantitative RT-qPCR The 2 delta delta Ct method is not the most accurate to analyze RT-qPCR data. The authors should have used the efficiency delta Ct method. For that, the authors should have first determine the efficiency of their respective pairs of primers using standard curves and use these efficiency values to calculate way more accurately relative gene expression (see Rao X, Huang X, Zhou Z, Lin X. An improvement of the 2ˆ(-delta delta CT) method for quantitative real-time polymerase chain reaction data analysis. Biostat Bioinforma Biomath. 2013 Aug;3(3):71-85. PMID: 25558171; PMCID: PMC4280562.)

Response: Thank you for this suggestion. We optimized the best reaction conditions for real time PCR by optimizing the Ta temperature (50-57°C) and different amounts of cDNA template (5-50ng/reaction) using a standard curve. We found the Ta=54°C and loading cDNA for each reaction (15ng/reaction) in 20μl showed high efficiency, which were the conditions we used for the CT analysis. The reference you suggested has now been cited.

  1. Minor comments -

Introduction:

  1. 69: "hos", please replace by "host" (t is missing).

Response: Thanks, the word has been modified.

References

  1. Liu, S.; Cai, P.; Hou, N.; Piao, X.; Wang, H.; Hung, T.; Chen, Q., Genome-wide identification and characterization of a panel of house-keeping genes in Schistosoma japonicum. Mol Biochem Parasitol 2012, 182 (1-2), 75-82.
  2. Cai, P.; Piao, X.; Hou, N.; Liu, S.; Wang, H.; Chen, Q., Identification and characterization of argonaute protein, Ago2 and its associated small RNAs in Schistosoma japonicum. PLoS Negl Trop Dis 2012, 6 (7), e1745.
  3. Cai, P.; Liu, S.; Piao, X.; Hou, N.; Gobert, G. N.; McManus, D. P.; Chen, Q., Comprehensive Transcriptome Analysis of Sex-Biased Expressed Genes Reveals Discrete Biological and Physiological Features of Male and Female Schistosoma japonicum. PLoS Negl Trop Dis 2016, 10 (4), e0004684
  4. Zhou, X. H.; Wu, J. Y.; Huang, X. Q.; Kunnon, S. P.; Zhu, X. Q.; Chen, X. G.,Identification and characterization of Schistosoma japonicum Sjp40, a potential antigen candidate for the early diagnosis of schistosomiasis. Diagnostic microbiology and infectious disease 2010, 67 (4), 337-45.

Round 2

Reviewer 1 Report

Some digital revisions made, but not the relevant one. Most important is whether the Histag from the Hsps was removed or not since these proteins expressed in E.coli. The Histag can trigger the T-cell response, therefore Figs. 6 and 7 were not improved and can produce an background or the response itself, since the data in Fig. 6 claim that the trigger is much better than the original SEA. A tagged control protein will help or otherwise the removal of the tag.

Round 3

Reviewer 1 Report

Dear Authors,

In principle the cited references say nothing about the relationship of target and inflammatory process as suggested.
Ref 1 says nothing about His6 and immune reactivity, no data about this point provided
Ref2 says The 6×His affinity-tag is poorly immunogenic; therefore, it is usually not necessary to remove the tag for the purposes of antibody generation, but this sentence is not documented by data.
Ref 3 is an experiment about a different target and improved the immune reactivity against Histag - one result

Author Response

Response:

Thank you for the comments. To answer the question raised by the reviewer 1, we previously cited 4 references (#1, 2, 3 and 4) to explain why we did not remove the His tag due to its low immunogenicity, which has been broadly accepted. 

In the Ref# 4 we provided before, Freire, T.; Lo-Man, R.; Bay, S.; Leclerc, C., Tn glycosylation of the MUC6 protein modulates its immunogenicity and promotes the induction of Th17-biased T cell responses. The Journal of biological chemistry 2011, 286, (10), 7797-811.

Authors clearly demonstrated that there are no specific responses of the lymph node cells, characterized by both lymphocyte proliferation and the introduction of IFN-γ when stimulated with the 6xHis protein.

In additional, in our experiment we also found that that rSjP40 can induce CD90.2+CD4- IL4 and CD90.2-CD4- IL4 responses in hepatic immune cells, which was not observed in these cells when stimulated by rSjP90 α (Fig 6). This evidence further highlighted the His tag portion of the recombinant proteins did not generate an IL4 response.

In ref#1, 2, 3, 5 as shown below authors all believed that His tag has low immunogenicity and used this point through their experiments.

In ref#1, Zhao, X.; Li, G.; Liang, S., Several affinity tags commonly used in chromatographic purification. Journal of analytical methods in chemistry 2013, 2013, 581093;

Authors mentioned several times about His-Tag, such as “The 6× His-tag has several merits, including a smaller size, absence of electric charge, low levels of toxicity, and immunogenicity [3].”

“The small-size tags (e.g., 6× His, FLAG, Strep II, and CBP) have the benefits of minimizing the effect on the structure, activity, and characteristics of the recombinant protein, and therefore usually there is no need to remove.”

In ref # 2, Loughran, S. T.; Walls, D., Purification of poly-histidine-tagged proteins. Methods Mol Biol 2011, 681, 311-35;

Authors described that “The 6×His affinity-tag is poorly immunogenic; therefore, it is usually not necessary to remove the tag for the purposes of antibody generation. In addition, in most cases, the 6×His tag does not interfere with the structure or function of the purified protein as demonstrated for a wide variety of proteins, including enzymes, transcription factors, and vaccines.”

In ref # 3, Ghose, C.; Eugenis, I.; Sun, X.; Edwards, A. N.; McBride, S. M.; Pride, D. T.; Kelly, C. P.; Ho, D. D., Immunogenicity and protective efficacy of recombinant Clostridium difficile flagellar protein FliC. Emerging microbes & infections 2016, 5, e8.

Authors described that “The His-tag was not removed following purification, given the low immunogenicity of such tags [34, 35]”

“Additionally, in the FliC-specific ELISA, cross-reactivity due to the presence of the low-immunogenic His-tag on the immunogen (TcdARBD and TcdBRDB) and the coating antigen (FliC) was not observed.”

Another reference (ref #5) also showed that 6×His is not necessary to be removed due to its low-immunogenicity.

Ref #5, D. Walls and S. T. Loughran, “Tagging recombinant proteins to enhance solubility and aid purification,” Methods in Molecular Biology, vol. 681, pp. 151–175, 2011.

Authors described that “The His-tag combines the advantages of being inert, of low-immunogenicity, and of small size (0.84 kDa) [80, 86]. In most cases, its small size means the His-tag does not interfere with the biochemical activities of the partner protein [87 – 93] or with most downstream applications [80].